# MYC Oncogene Contributions to Release of Cell Cycle Brakes

**DOI:** 10.3390/genes10030244

**Published:** 2019-03-22

**Authors:** Lucía García-Gutiérrez, María Dolores Delgado, Javier León

**Affiliations:** 1Instituto de Biomedicina y Biotecnología de Cantabria (IBBTEC) CSIC-Universidad de Cantabria and Department of Biología Molecular, Universidad de Cantabria, 39011 Santander, Spain; lucia.garcia@ucd.ie (L.G.-G.); maria.delgado@unican.es (M.D.D.); 2Systems Biology Ireland, University College Dublin, Belfield, Dublin 4, Ireland

**Keywords:** MYC, cell cycle, CDK inhibitors, p21, p27, p15, ARF

## Abstract

Promotion of the cell cycle is a major oncogenic mechanism of the oncogene c-MYC (MYC). MYC promotes the cell cycle by not only activating or inducing cyclins and CDKs but also through the downregulation or the impairment of the activity of a set of proteins that act as cell-cycle brakes. This review is focused on the role of MYC as a cell-cycle brake releaser i.e., how MYC stimulates the cell cycle mainly through the functional inactivation of cell cycle inhibitors. MYC antagonizes the activities and/or the expression levels of p15, ARF, p21, and p27. The mechanism involved differs for each protein. p15 (encoded by *CDKN2B*) and p21 (*CDKN1A*) are repressed by MYC at the transcriptional level. In contrast, MYC activates ARF, which contributes to the apoptosis induced by high MYC levels. At least in some cells types, MYC inhibits the transcription of the p27 gene (*CDKN1B*) but also enhances p27’s degradation through the upregulation of components of ubiquitin ligases complexes. The effect of MYC on cell-cycle brakes also opens the possibility of antitumoral therapies based on synthetic lethal interactions involving MYC and CDKs, for which a series of inhibitors are being developed and tested in clinical trials.

## 1. Introduction

The oncogene c-MYC (referred to herein as MYC) was the first described gene that encoded for an oncogenic transcription factor with the ability to transform cells in culture. MYC is overexpressed by different mechanisms in 60–70% of human solid and hematopoietic tumors [1,2,3,4,5]. The MYC family of proteins is composed of three members: c-MYC, N-MYC, and L-MYC. The existence of multiple MYC family members with distinct expression patterns reflects different requirements of MYC during development and in the adult animal, which is consistent with the specific way each gene is deregulated in certain cancer types [6].

MYC is a transcription factor of the helix-loop-helix-leucine zipper (HLH-LZ) family that regulates the activation or repression of many target genes [7,8]. Regulation of transcription by MYC depends on the formation of heterodimeric complexes with MAX protein [9]. The MYC-MAX heterodimer is the active form, which binds to specific DNA sequences called E-boxes (canonical sequence CACGTG) in the regulatory regions of target genes. The MYC network (also known as the MAX-MLX network), includes other components of the HLH-LZ family such as the MXDs, MNT, MLX and others, with different functions in gene expression regulation upon binding to E-boxes in the DNA (for recent reviews see [10,11]).

The number of MYC-binding sites revealed by genome-wide technologies ranks between 7000 and 15,000 in different models. Indeed, MYC is bound at one or more sites of the regulatory regions of 10–15% of human genes [6,7,8,12]. In agreement with the large number of MYC target genes, overexpression of MYC deregulates a series of biological functions such as cell-cycle progression, nucleotide biosynthesis, energy metabolism, protein synthesis and ribosome genesis, genomic maintenance, immortalization, and differentiation [1,7,13,14,15]. Such deregulation confers ample competitive advantages to the cell and contributes to the well-stablished role of MYC in a wide variety of cancers.

MYC protein contains several domains that play important roles in MYC functions, as well as many residues susceptible of being modified, modulating MYC’s activity and stability [6]. MYC contains an unstructured N-terminal region, which includes two conserved regions known as MYC boxes (MB) (Figure 1). MBI and MBII are located within the transcriptional transactivation domain (TAD), essential for MYC transcriptional and cell-transforming activity. The MBII is crucial for the recruitment of MYC transactivation co-activators such as TRRAP, GCN5, TIP48, TIP49, TIP60, CBP/p300, as well as SKP2 [16,17,18]. The central region of MYC contains the MBIII, which has been shown to be important for transcriptional repression [19,20] and MBIV, needed for MYC transcriptional activity and MYC induced apoptosis [21]. The C-terminal region of MYC includes the basic, helix-loop-helix, and leucine zipper domains (b-HLH-LZ). Through the basic domain, MYC protein recognizes specific sequences and binds the DNA, while the HLH-LZ domain mediates the dimerization with its major partner MAX [9,22,23].

Activation or repression of MYC-regulated genes is mediated by its interaction with a variety of partner proteins, many of them involved in chromatin structure regulation (recently reviewed in [17,24]). The mechanism for MYC-mediated transactivation depends on the recruitment of complexes containing histone acetyltransferases (HATs) [7,8] (Figure 1a). TRRAP (Transformation-Transactivation domain Associated Protein) was originally isolated as a cofactor of MYC and recruited to most of the MYC target genes upon mitogen stimulation [25,26]. Two different TRRAP containing complexes possess GCN5 HAT activity. TRRAP-containing TIP60 complex consists of the TIP60 HAT, the ATPase/helicase motif containing cofactors TIP48 and TIP49 and the SWI/SNF related protein p400 ATPase. Both GCN5 and TIP60 acetylate histones at MYC target genes. Furthermore, CBP/p300 interacts with MYC mediating its acetylation, increasing MYC stability and stimulating MYC-transcriptional activation [17]. MYC is present at the promoter of nearly all active genes acting as an amplifier of the transcription already going on at those genes [27,28] although there is some selectivity on the genes regulated by MYC [29,30]. Different studies support the idea of MYC as a transcription amplifier because of its role regulating global transcriptional pause release [31]. The mechanism is not well known but the activating interaction of MYC with P-TEFb (positive transcription elongation factor b) likely plays an important role in it [32] (Figure 1a).

Apart from transcriptional activation of gene expression, MYC also represses a great number of genes, many of them involved in processes such as the inhibition of cell-cycle progression and cell adhesion [33,34]. MYC represses transcription by interacting with other transcription factors and co-repressor complexes at the core promoter region of genes. So far, MYC has been reported to exert its repression activity by interacting mainly with SP1 and/or MIZ-1 (Figure 1b). These two transcription factors normally activate transcription. However, interaction with MYC switches them into transcriptional repressors mainly by displacing SP1 and MIZ-1 co-activators. For example, MIZ-1 recruitment of p300 can be antagonized by MYC [35,36]. Further, MYC represses transcription through SP1 by recruiting histone deacetylases (HDACs) [37]. SP1-SMAD complex has been found to be inactivated by MYC resulting in gene repression [38]. MYC also interacts with SIN3 [19] and with HDAC3 [20]. In this way, MYC recruits HDACs to the core promoter of several genes, resulting in transcriptional repression. The MYC-MIZ-1 complex can recruit the DNA methyltransferase DMNT3A to promoters, repressing transcription. This might be an efficient mechanism to repress CpG island promoters [39]. At least two of the genes known to be repressed by MYC through these mechanisms encode proteins involved in cell-cycle regulation: *CDKN1A* (p21^CIP1^) [40,41], *CDKN2B* (p15^INK4B^) [35,38,42].

We will review here the role of MYC as cell-cycle brake releaser i.e., how MYC stimulates cell cycle mainly through the repression of cell-cycle inhibitors (Figure 2). Cell-cycle progression is regulated by serine/threonine protein kinases composed by a catalytic subunit or CDK (cyclin-dependent protein kinase), and a regulatory subunit, the cyclin [43,44]. CDK1, 2, 4, and 6 and A, B, E, and D-type cyclins constitute the major regulators of the mammalian cell cycle. D-type cyclins (D1, D2, and D3) preferentially bind and activate CDK4 and CDK6 at early G_1_-phase of the cell cycle, leading to the phosphorylation of the retinoblastoma protein (RB) and the release of the E2F transcription factors [45,46]. Cyclin E1/2-CDK2 complexes in the late G_1_-phase further phosphorylate RB, allowing the expression of E2F target genes required for the transition to S-phase [47]. Later, CDK2 complexes with Cyclin A2. Cyclin A is required for DNA replication and is expressed through S and G_2_ phases. M-phase transition is regulated by CDK1 activated by B-type cyclins (B1 and B2) [43,48]. CDK inhibitory proteins (CKIs) accomplish an additional level of regulation of the cell cycle. CKIs are divided into two families (Figure 2). The INK4 family (consisting of p16^INK4A^, p15^INK4B^, p18^INK4C^, and p19^INK4D^) binds and inhibits CDK4 and CDK6 kinases, impairing their association with D-type cyclins. The CIP/KIP family (consisting of p21^CIP1^, p27^KIP1^, and p57^KIP2^) inhibits progression at every cell-cycle phase upon binding to several already formed Cyclin-CDK complexes [49]. CDK inhibitors are involved in the regulation of a variety of biological processes beyond cell-cycle regulation [50] and some of them play important roles in cancer [51].

MYC stimulates cell-cycle progression through the regulation of many genes related to cell-cycle control (recently reviewed in [13]) (Figure 2). MYC induces critical positive cell-cycle regulators such as cyclins (D-type cyclins, E-type cyclins, cyclin A and cyclin B1), CDKs (CDK1, 2, 4, 6), and E2F transcription factors (E2F1, 2, 3) (reviewed in [13]). Moreover, MYC antagonizes the activity of cell-cycle inhibitors such as p15, p21, and p27 by different mechanisms. These activities of MYC will be discussed below.

## 2. MYC and the *INK4A*/*ARF*/*INK4B* Locus

The *INK4A*/*ARF*/*INK4B* gene locus is located on chromosome 9p21 in humans encoding three related proteins: p15^INK4B^ (p15 herein after), p14^ARF^ in humans or p19^ARF^ in mice (ARF herein after) and p16^INK4A^ (p16 herein after). p15 and p16 are characterized for their direct interaction with CDK4 and CDK6, blocking the formation of cyclin D-CDK4/6 complexes and provoking arrested proliferation through preventing phosphorylation of RB and S-phase entry [52]. On the other hand, ARF protein is unrelated with the INK4 family of CDK inhibitors but it shares the exons 2 and 3 with p16^INK4A^ gene, while the first exon of each gene is totally different. They are transcribed from an alternative reading frame (i.e., ARF) within the same locus and thus, their amino acid sequences lack any similarity. ARF induces cell-cycle arrest in G_1_ and G_2_ phases [53] and/or apoptosis through the regulation of the ARF/MDM2/p53 apoptotic pathway mainly, although induction of p53-independent apoptosis has also been reported to be mediated by ARF [54,55]. Albeit activation of the p53 apoptotic pathway is commonly mediated by DNA damage or cellular stress responses, ARF acts as an unusual tumor suppressor, being activated by oncogenic signals such as MYC [56] among others (reviewed in [57]). This response is considered as a security measure to avoid aberrant and uncontrolled proliferation due to sustained growth signaling. In fact, the expression of the *INK4A*/*ARF*/*INK4B* locus is lost in a wide range of human tumors (reviewed in [58]). Disruption of the exon 2 of *INK4A* makes mice more prone to tumor development, an alteration that affects both p16 and ARF. However, specific deletion of the ARF exon 1 in mice lead to the same phenotype while harboring intact p16, confirming ARF as a tumor suppressor playing a key role in protecting cells from aberrant proliferation [59]. In agreement, immortalization of primary mouse embryonic fibroblasts (MEFs) implies normally loss of either ARF or p53 [60,61] and MYC can immortalize MEFs [62,63] through a process that is normally accompanied by either ARF or p53 loss of function [56]. Here we will review the ARF-regulation by MYC and vice-versa, as ARF controls MYC’s activity to prevent abnormal proliferation and oncogenic transformation.

### 2.1. MYC and p15^INK4B^ Regulation

The cell-cycle inhibitor p15 arrests proliferation in G_1_ phase by specifically inhibiting cyclin D-CDK4/6 complexes [64]. Moreover, high levels of p15 redistribute p27 from cyclin D-CDK4/6 complexes to cyclin E-CDK2 complexes, leading to arrested proliferation [65]. Treatment of lung epithelial cells with TGFβ lead to a rapid downregulation of MYC levels, while p15 was highly induced. However, exogenous MYC expression resulted in the inhibition of TGFβ-mediated p15 induction [66]. In fact, MYC inhibits the activation of a reporter gene under the control of the proximal region of p15 promoter. This region contained the TGFβ responsive element (TGFβ-RE) and the transcriptional initiator site (Inr) [66]. The repression of p15 by MYC occurs through either mechanisms that involve or not the Inr element. The Inr element consists of a weak consensus sequence located at the transcription start site (TSS) of different promoters through which MYC is known to exert part of its repression activity (reviewed in [67,68]). Different proteins have been described to cooperate with MYC in the binding to the Inr element, such as YY1, TFII-I and MIZ-1 [69,70,71]. The zinc-finger protein MIZ-1 recognizes and binds the Inr element of its target genes promoting their activation, such as *INK4B* upon TGFβ treatment. MYC-MAX heterodimers impair *INK4B* expression by interacting with MIZ-1 at the Inr element of its promoter, preventing p300 recruitment by MIZ-1 [35]. TGFβ inhibited the interaction of MYC with MIZ-1, leading to *INK4B* induction by MIZ-1 through its interaction with SMAD proteins [42]. On the other hand, MYC can repress *INK4B* expression independently of the Inr element. This mechanism involves the interaction of MYC with SP1 and SMAD proteins. MYC binds to activated SMAD, forming a repressor complex together with SP1, leading to the inactivation of *INK4B* expression upon TGFβ treatment [38].

### 2.2. MYC Regulation of ARF Expression

Although MYC is always related to enhanced proliferation and cell growth, deregulated MYC expression paradoxically triggers apoptosis upon cellular stress conditions such as serum deprivation [56,72]. This process takes places mainly through the 53-dependent apoptosis pathway [73,74] although it has been reported to also happen in a p53-independent manner [75] in some cell types. Thus, cells overexpressing MYC are subjected to a high selection pressure to proliferate in the absence of growing factors, in which programmed cell death mechanisms need to be abrogated. MYC-induced apoptosis is mainly mediated by the induction of ARF expression at the mRNA level, leading to the inactivation of MDM2 by its sequestration to the nucleolus and thus, stabilization and activation of p53. Activation of p53 results in subsequent induction of p21 and other proteins involved in the p53-dependent apoptosis pathway [76]. In fact, p53-null cells showed resistance to MYC-induced apoptosis, while the effect observed in ARF-null cells was less compromised [56]. MYC has been found to induce p53 expression in an ARF-independent manner, although p53-dependent apoptosis was significantly compromised in ARF-null cells [56]. Furthermore, lymphomagenesis induced by MYC in Eµ-*MYC* transgenic mice [77], selectively inactivates either ARF or p53 in most tumors, being both genes found mutated with similar frequency [78]. In agreement with previous results obtained in MEFs, Eµ-MYC derived pre-B cells showed high rates of apoptosis and increased ARF levels, while p53 levels remained constant when compared to control cells. Thus, high rates of spontaneous cell death in this model correlated with ARF activation [78]. Although in most of the cases, ARF and p16 were inactivated due to mutations within their shared DNA sequences, retained expression of non-altered p16 found in some of these tumors brought to light the importance of ARF but not p16 for B-cell lymphoma development [78]. Thus, loss of ARF attenuates MYC-induced apoptosis in vivo, allowing prevalence of MYC oncogenic activity leading to high rates of tumor formation. In agreement, *INK4A*/*ARF*^−/−^-Eµ-MYC mice were more prone to develop lymphomas and displayed apoptotic defects despite the presence of wild-type p53, a phenotype similar to the one observed in p53-null lymphomas [79]. Other studies using mouse models with restricted expression of the oncogene MYC to the epidermis and other epithelial tissues [80,81] showed nearly completely abrogated apoptosis in a p53-null background [81] and highly reduced in ARF-null mice [82], consistent with previous studies. Moreover, ARF modulated specifically MYC-mediated apoptosis, while MYC-mediated stimulation of proliferation was not affected in the absence of ARF in the epidermis.

The mechanism of ARF expression induction by MYC remains largely unclear, although it seems to happen through an indirect mechanism involving the regulation of other factors that directly activate ARF expression. MYC induces FoxO transcription factors, which bind to and regulate the *INK4A*/*ARF* locus activating ARF expression. Thus, constitutive MYC signaling induces both nuclear FoxO levels and ARF expression [83]. On the other hand, the transcription factor E2F1 directly induces ARF [84], although this pathway does not seem to be conserved in mouse [85]. As MYC is known to directly regulate E2F1 expression, MYC-mediated ARF upregulation through E2F1 regulation has been suggested [86]. On the other hand, MYC has been reported to modulate ARF protein stability by interfering with ARF ubiquitination and degradation. ARF is very unstable in normal cells, while its degradation is inhibited in cancerous cells. The ubiquitin ligase ULF has been reported to ubiquitylate ARF leading to its degradation in vitro and in vivo. Furthermore, MYC can interact with ULF, impeding ARF ubiquitination and thus, increasing its stability [87]. This control of ARF stability is thought to be a mechanism by which the cell senses and distinguishes between normal versus overexpressed MYC. Thus, only upon oncogenic MYC levels, ULF-mediated ARF degradation is inhibited and therefore the apoptotic response is activated [88]. Consistently, physiological levels of MYC did not activate ARF promoter [89].

### 2.3. ARF-Mediated Regulation of MYC Activity

Apart from the p53-dependent ARF induction of apoptosis and arrested proliferation through MDM2 sequestration, ARF has been proposed to have p53- and MDM2-independent functions to suppress cell proliferation [75]. Moreover, ARF has been suggested to interact with targets other than p53 and MDM2 to inhibit proliferation [54]. ARF was found to interact with MYC to relocalize it from the nucleoplasm to the nucleolus and thus, inhibiting MYC-activated transcription and leading to G_1_ arrest in a p53-independent manner [90]. An ARF mutant lacking the N-terminal domain of the protein failed to interact and colocalize with MYC and thus, was not able to inhibit MYC-activated transcription [90]. In contrast, other studies have shown that upon ectopic MYC expression, ARF is relocalized from the nucleolus to the nucleoplasm and colocalized with it. The same result was obtained upon MYC-ER activation, a chimeric protein consisting of MYC fused to the estrogen receptor and activatable by 4-hydroxy-tamoxifen. [91]. This discrepancy has been attributed most likely to the different systems used for each study and the different ratio levels between ARF and MYC in each model. Thus, MYC/ARF localization is bidirectional. MYC interacts with ARF through two different domains, one through the TAD situated at the N-terminal, and the other one through the HLH-LZ domain, located at the C-terminus of MYC [17]. Although the C-terminal domain had only a minimal effect over ARF interaction when deleted, depletion of the TAD completely abrogated MYC-ARF interaction [91]. Notably, ARF antagonizes the SKP2-mediated ubiquitylation of the MYC TAD [92]. MYC-p14ARF interaction has also been demonstrated and takes place through the MBII of MYC. This interaction leads to inhibition of MYC-induced transcription and nucleolar localization of MYC [93]. Chromatin immunoprecipitation assays showed that ARF was recruited to active MYC target genes, forming complexes with MYC-MAX heterodimers, impairing MYC-transactivating activity without affecting MYC-transrepressing activity [57]. Thus, this mechanism of ARF blocking MYC transactivation of genes impairs MYC-mediated hyperproliferation probably by ARF-mediated interference of TAD interaction with MYC-coactivators [91]. Many target genes which are repressed by MYC are involved in anti-apoptotic functions. The fact that ARF impairs MYC transactivation activity but that it does not interfere with MYC repression mechanisms would favor the pro-apoptotic response within the cells upon deregulated MYC activity [94,95,96].

## 3. MYC and p21 Regulation

The CIP/KIP cell-cycle inhibitor p21^Cip1/Waf1^ (p21), encoded by the *CDKN1A* gene, plays key roles in controlling cellular processes such as proliferation, senescence, cell differentiation and apoptosis (reviewed in [97,98]). Similar to its relative p27, p21 interacts with cyclin-CDK complexes inhibiting cell-cycle progression [99,100] in response to different stimuli. p21 is a transcriptional target of p53, essential for p53 induced cell-cycle arrest in G_1_ and G_2_ phases upon DNA damage [101,102]. One of the first evidences in which MYC was found to have an opposite effect over p21-mediated cell-cycle arrest was reported by Perez-Roger and colleagues, when they showed that MYC promoted p21 sequestration through induction of D-type cyclins [103]. While a strong RAF signal was found to promote cell-cycle arrest through p21 induction in NIH 3T3-derived cells [104], MYC-ER activation was able to counteract this effect by an increase in cyclin D2-p21 binding that was proportional to the increase in cyclin D2 expression mediated by MYC [103]. However, MYC-ER activation did not lead to increased cyclin D1 expression in this system, in agreement with the lack of increased binding of p21 to cyclin D1 upon these conditions [103]. One of the major mechanisms by which MYC induces S-phase entry relies on MYC’s ability to activate cyclin E-CDK2 complexes [13]. Thus, apart from the induction of Cyclin E expression (among others), MYC-mediated release of cyclin E-CDK2 inhibition though induction of cyclin D2 and further sequestration of p21 in cyclin D-CDK4/6 complexes [13] constitutes a remarkably important process in MYC’s role as pro-proliferative agent.

### 3.1. MYC-Mediated p21 Repression by Direct Recruitment to Its Core Promoter Region

The better characterized and most studied mechanism by which MYC is known to counteract the antiproliferative activity of p21 occurs at the transcriptional level (Figure 3). In fact, p21 has been reported to be one of the major targets of MYC repression [105]. This regulation of p21 by MYC is a clear example of MYC as transcriptional repressor, an idea that is becoming widely accepted and studied and that seems to account for at least half of MYC’s activity as transcriptional regulator, as revealed in transcriptomic analysis upon MYC enforced expression.

Several mechanisms have been reported as per which MYC is able to repress transcription (reviewed in [34,36]), however further research needs to be performed to better understand how this process takes place. Histone deacetylase recruitment to promoter regions is a well-known mechanism of transcriptional repression. Indeed, trichostatine A (a histone deacetylase inhibitor) treatment has been shown to induce p21 expression [106]. Different studies have found that MYC-mediated *CDKN1A* transcriptional repression occurs in a HDAC-independent manner [40,107]. Besides, cells stably expressing the MYC-ER construct repressed the expression of *CDKN1A* upon MYC-ER activation, even in the absence of de novo protein synthesis. The inhibition of de novo protein synthesis diminishes the possibility that an intermediate protein could be responsible for this effect, meaning that MYC directly triggers p21 repression [40]. The *CDKN1A* promoter contains three non-canonical E-box sequences, two of them close to the transcription start site (TSS) (−5 to +1 bp and −20 to −15 bp) and another one around 150 bp upstream the TSS (−162 to −157 bp) (Figure 3a). Whether MYC repression activity relies on MYC’s ability to recognize and interact with the DNA through E-boxes is not yet determined. In the case of *CDKN1A*, direct MYC DNA binding has not been reported so far, thus its activity on *CDKN1A* promoter is E-box independent. Different studies reported that a short sequence within the transcription start site (from around −150 to +16 bp) is enough for MYC to repress *CDKN1A* promoter’s activity [40,41,107]. This promoter region contains several responsive elements as shown in Figure 3. MYC is recruited to the promoter DNA sequence by interacting with other transcription factors involved the regulation of *CDKN1A* expression, being SP1/SP3 and MIZ-1 the main ones described so far [40,41,108]. TGFβ treatment of murine and human keratinocytes leads to MYC downregulation followed by p21 induction and cell-cycle arrest [107]. Luciferase assays using different *CDKN1A* promoter fragments revealed that the TGFβ responsive element is not needed for MYC-mediated p21 repression. A luciferase construct containing from −62 to +16 bp of the *CDKN1A* promoter, was enough for MYC to mediate promoter repression and thus, MYC exerts its regulation independently of the rest of elements that act upstream that sequence, such as p53 or C/EBP [40]. Within the vicinity of the *CDKN1A* transcription start site that is enough for MYC to repress p21 expression, there are multiple SP1 binding sites and a potential Inr sequence. The initiator binding protein (TFII-I) induces gene transcription from the Inr of certain TSS and MYC is known to interact with the TFII-I impeding its activity in other models. However, that was not the case for *CDKN1A*, as depletion of the Inr sequence (+7 to +16 bp from the TSS) did not affect MYC repression of *CDKN1A* promoter in colorectal adenocarcinoma cells. Instead, MYC was found to interact with SP1 and SP3 transcription factors which play important roles in the induction of p21 expression [108]. The central part of the MYC protein, from amino acids 143 to 352, is essential for MYC to interact with the zinc finger domain of SP1 and enough to counteract SP1 induction of *CDKN1A* expression in CaCo cells [40].

The mechanism by which MYC represses *CDKN1A* promoter activity seems to be cell-type dependent. MYC also represses *CDKN1A* expression by interacting with the initiator-binding transcription factor MIZ-1. During hematopoietic differentiation, MIZ-1 levels increase and trigger *CDKN1A* expression, while ectopic MYC expression repressed basal or TPA-induced *CDKN1A* levels [41]. The MYC responsive region of *CDKN1A* promoter in this model was found to be between −49 and +16 bp from the transcription start site, a sequence already reported in other studies, as mentioned above. Nevertheless, opposite to previous reports [40], the Inr sequence was essential for MIZ-1-dependent recruitment of MYC to impair *CDKN1A* expression in other studies [41]. Again, MYC binding to the DNA was not necessary as the basic domain of MYC protein is not needed for *CDKN1A* repression. Instead, MYC was recruited to the DNA by interacting through its HLH domain with MIZ-1 [41,109]. The MYC^V394D^ mutant (HLH mutated domain), unable to interact efficiently with MIZ-1 although still capable of interacting with MAX, allowed p21 expression and cell differentiation, bringing to light that MYC-MIZ-1 interaction is essential for *CDKN1A* repression [41].

More recently, MYC has been shown to form a ternary complex with MIZ-1 and GFI-1 able to bind the *CDKN1A* core promoter resulting in p21 repression [110] (Figure 3b). GFI-1 is a nuclear transcriptional repressor found to have important roles in hematopoietic cells [111,112,113] as well as in other tissues [114,115,116] and it has been reported to cooperate with MYC in lymphomagenesis [117,118]. GFI-1 regulates *CDKN1A* expression by recruitment of HDAC1 and G9a [119,120]. Nevertheless, although GFI-1 has two binding sites located 1.4 and 2.8 Kb upstream *CDKN1A* TSS, GFI-1 repression of *CDKN1A* expression happened through a mechanism that is independent of its DNA binding ability [119,120]. Instead, and according to this study [110], recruitment of both MYC and GFI-1 is dependent on MIZ-1 leading to the formation of a ternary complex that binds *CDKN1A* core promoter. Knocking down MIZ-1 expression leads to a significant decrease in MYC and GFI-1 occupancy at the *CDKN1A* promoter region. Indeed, MIZ-1 binds GFI-1 through its C-terminal 1-12 ZFs, while the regions flaking the ZFs are required for MYC interaction [109]. Besides, TGFβ not only would induce p21 through reduction of MYC expression [107], but also it reduces the levels of GFI-1, an effect that may contribute to the disruption of the MIZ-1/MYC/GFI-1 complex at the *CDKN1A* promoter region allowing p21 expression [110].

Another ternary complex involving MYC and MIZ-1 together with DMNT3A has been described to inhibit *CDKN1A* expression by inducing CpG methylation within the *CDKN1A* core promoter (Figure 3b) [39]. Combination of ectopic expression of MYC and DMNT3A has been found to highly repress *CDKN1A*, while downregulation of DMNT3A restores its expression [39]. MYC recruits DMNT3A to the core promoter of *CDKN1A* through MIZ-1, forming a ternary complex in which MYC is essential for bringing together MIZ-1 and DMNT3A [39]. Moreover, inhibition of DNA methyltransferase activity through 5-aza-cytidine abolished the MYC-mediated repression of *CDKN1A*, proving that DNA methyltransferase activity is needed for MYC to accomplish p21 downregulation [39]. On the other hand, histone demethylation activity has been reported to cooperate with MYC in *CDKN1A* repression. MYC forms a complex with TFAP2C (AP2C herein after) and the histone demethylase KDM5B capable to bind and repress the core promoter of *CDKN1A* through the AP2-binding site located −111 to −103 bp from the TSS (Figure 3b). Although AP2C and MYC are capable of repressing p21 expression alone, recruitment of KDM5B is dependent on both transcription factors and required for an optimal *CDKN1A* repression [121]. Thus, MYC would not only block the expression of p21 by interfering with factors that upregulate its expression, it will also actively modulate *CDKN1A* transcription by recruitment of DNA methyltransferase and histone demethylase activities to its core promoter.

### 3.2. MYC-Dependent Switch from Cell-Cycle Arrest to Apoptosis by Inhibiting p53-Dependent Activation of p21 Expression

Activation of the p53 pathway upon DNA damage can lead to two different outcomes, either cell-cycle arrest, mediated by the p53-direct induction of *CDKN1A* transcription, or apoptosis, mediated by p53 induction of *PUMA* and *PIG3*, among other target genes. MYC plays a very important role in the choice of this response. By interacting with MIZ-1, MYC is recruited to the proximal promoter region of *CDKN1A* leading to the inhibition of p53 mediated p21 expression in HTC116 cells upon MYC overexpression [122]. MYC did not affect p53 binding to *CDKN1A* promoter neither that of *PUMA* or *PIG3*, but specifically inhibited p21 expression promoting PUMA-mediated apoptosis instead of p21-dependent cell-cycle arrest [123]. Similar results were obtained in K562 cells, in which p53 activation lead to apoptosis or cell-cycle arrest while MYC overexpression significantly impaired apoptosis and p21 induction by p53, without affecting BAX expression [124].

### 3.3. MYC-Mediated Inhibition of RAS-Induced CDKN1A Expression

Cooperation between RAS and MYC in cellular transformation was the first example of oncogenes cooperation and has been widely studied since then [62]. Apart from its pro-proliferative activity, RAS is known to induce cell-cycle arrest and senescence in different models of primary cells [125,126] and chronic myeloid leukemia (CML) cells [127]. This mechanism of RAS-induced cell-cycle arrest involves the induction of cell-cycle inhibitors such as p16 (leading to RB inactivation), ARF and p53 that subsequently activates p21 expression and cell-cycle arrest. The mechanism through which RAS mediates p21 induction was first described to happen mainly through SP1 sites 2 and 4 in Cos7 cells [128]. Few years later, RAS induction of p21 expression was reported to be dependent on RAF in a model of CML (K562 cells). In this study, the SP1 sites 2 and 5 are the ones that account for the main RAS transactivation activity on *CDKN1A* promoter, although sites 3 and 4 also contributed to it [108]. Like other models already described, MYC was able to impair RAS-induced *CDKN1A* expression by binding to SP1 and inhibiting SP1-mediated *CDKN1A* expression regardless of the SP1 site analyzed [108] (Figure 3b). HLH and MB2 domains were needed for MYC to exert its repression on *CDNK1A* promoter upon RAS activation, in a process independent of MIZ-1 [108]. Thus, MYC exerts a major role in controlling the *CDKN1A* promoter in a silent state in CML, promoting cell-cycle progression and contributing to tumorigenesis. However, in agreement with the fact that cell context is essential to determine the outcome of a biological process, depending on the signal that induces the expression of p21, MYC will mediate its repressive activity over *CDKN1A* promoter through one mechanism or another. In fact, MYC seems to adapt its regulation ability according to the factor which mainly regulates p21 expression depending on the cellular context. These multiple mechanisms of MYC-induced p21 repression bring to light the importance of p21 regulation for MYC to promote cell proliferation and transformation.

### 3.4. MYC-Indirect Repression of CDKN1A Expression

Apart from the direct regulation of p21 transcription by MYC through its recruitment to the core promoter of the *CDKN1A* gene, mediated by protein-protein interactions with other *CDKN1A* regulators, MYC can induce transcription factors and miRNAs that are directly involved in the regulation *CDKN1A* expressions. TFAP4 (AP4 herein after) is a direct MYC target gene that belongs to the bHLH-LZ family of transcription factors. Its basic DNA-binding domain is essential to mediate *CDKN1A* repression through recognition of the E-boxes located at the core promoter of this gene [129,130]. AP4 only forms homodimers, so that it is very unlikely that AP4 exerts its repression by interacting with other transcription factors. Instead, it may compete for the occupancy of the E-boxes with other bHLH-LZ transcription factors known to induce *CDKN1A* expression [131]. AP4 is known to repress gene expression by recruitment of HDAC (HDAC1 and HDAC3) to core promoters [132,133]. Nevertheless, inhibition of HDAC activity is not enough to abolish AP4-mediated *CDKN1A* repression [134], in agreement with the HDAC-independent MYC-mediated p21 repression already addressed [40,107]. Other studies have described other potential mechanism for AP4-mediated p21 repression in which AP4 would impair TBP interaction with the TATA-box within the TSS, preventing the assembly of the RNA polymerase II complex [132,135].

Finally, MYC has been shown to regulate p21 expression at the post-transcriptional level, by modulation of miRNA expression (Figure 4). p21 is a major target of the miR-17 family of miRNAs and it has also been reported that silencing of p21 due to aberrant regulation of miRNA-17 contributes to tumorigenesis [136,137,138]. Moreover, the miR-17 family members correlate with MYC expression [139,140,141] and indeed, *miR-17-5p*, *miR-20a*, and *miR-106a*, all of them belonging to the miR-17 family of miRNAs are induced by MYC and downregulate p21 expression [142]. Thus, miRNA regulation by MYC indirectly regulates p21 expression contributing to the promotion of cell proliferation by MYC.

## 4. MYC and p27 Regulation

The cell-cycle inhibitor p27^Kip1^ (p27), encoded by the *CDKN1B* gene, is known to induce proliferation arrest in G_1_ by blocking the kinase activity of cyclin-CDK complexes, being cyclin E-CDK2 inhibition which exerts its main role in cell-cycle control. Besides, p27 behaves as a transcriptional regulator involved in a variety of cellular functions and in cancer (recently reviewed in [143]). Since MYC is a well-known potent inductor of the transition from G_1_ to S-phase, the antagonistic effect found between MYC and p27 in the control of cell-cycle progression has been a matter of study for many years. This is consistent with the fact that *MYC*^−/−^ cells showed increased levels of p27 and inhibition of cyclin-CDK activity, together with reduced proliferation rates [144]. Moreover, the opposite correlation between high levels of MYC and low levels of p27 has been found in many human tumors and it is considered as poor prognosis of the disease [145,146]. There are several mechanisms through which MYC counteracts p27 activity, thus enabling the G_1_-S transition: (i) repression of p27 at the transcriptional level; (ii) induction of *miR-221* and *miR-222* that down-regulate p27 expression; (iii) induction of D-type cyclins and CDK4 and CDK6 that sequester p27 from cyclin E-CDK2 complexes; (iv) induction of *CCNE* expression directly or through E2F, leading to activation of cyclin E-CDK2 complexes that antagonize p27 function; v) induction of different components of the SCF^SKP2^ ubiquitin ligase complex (i.e., *CKS1*, *CUL1* and *SKP2*) that targets p27 for proteasomal degradation (Figure 4). Mechanisms (i) and (ii) lead to *CDKN1B* regulation at the mRNA level, either due to promoter repression or post-transcriptional regulation. However, these two mechanisms of repression account for a minor percentage of MYC-mediated p27 regulation. The last three mechanisms contribute to p27 downregulation in a much higher extent, mostly involving p27 protein sequestration and degradation by MYC [147]. These mechanisms are discussed below.

### 4.1. Repression of CDKN1B Expression

One of the mechanisms that accounts for the inhibition of p27 by MYC involves MYC-mediated transcriptional repression of the *CDKN1B* core promoter, as already described for its related *CDKN1A* gene. *CDKN1B* mRNA expression levels inversely correlate with MYC expression in immune cells and other models. B cell receptor (BCR) engagement in immature B cells (upon anti-IgM treatment) lead to *MYC* downregulation, followed by p27 expression and induction of apoptosis [148], an effect that is reproduced upon siRNA-mediated MYC downregulation [149] and blocked by MYC [148,149,150]. Thus, there is an inverse correlation between *MYC* and *CDKN1B* mRNA expression levels in this model upon IgM treatment. Luciferase assays showed that the *CDKN1B* promoter region containing from −2002 to +154 bp responded to anti-IgM treatment leading to an increase in *CDKN1B* promoter activity [151]. The *CDKN1B* promoter contains an Inr element at the TSS which, as already described for other MYC-repressed target genes, has been found to be crucial for *CDKN1B* downregulation and MYC has been reported to interact with it in different models [151]. Indeed, *CDKN1B* upregulation upon BCR engagement is abrogated by ectopic MYC expression [151]. MYC interaction with the Inr element relies on its MBII and accordingly, a MYC^P115L^ mutant, in which the Phe at the position 155 is replaced by a Leu within the region of MYC needed for its transcriptional suppression function, enhances its repressor activity [151], consistently with other known MYC-repression mechanisms [67,152,153]. Later studies showed that MYC represses *CDKN1B* promoter by direct interaction and inhibition of Foxo3a, a transcription factor known to upregulate *CDKN1B* expression [154]. In fact, immature B cells subjected to anti-IgM treatment showed an increase in Foxo3a expression [155], which can be abrogated by MYC expression [154]. Opposite to what was found for *CDKN1A*, MYC interacts with the *CDKN1B* Inr element through MAX, blocking *CDKN1B* expression [151].

### 4.2. MYC-Induced Repression of p27 Through miRNA Up-Regulation

Regulation of *CDKN1B* at the post-transcriptional level by miRNAs has been recurrently reported during the last years. Indeed, aberrant up-regulation of miRNA clusters that regulate p27 expression have been linked with cancer development, progression and invasion [156,157,158], bringing to light the importance of p27 regulation at this level. Screening of the miRNAs involved in the regulation of p27 revealed that the *miR-221* family of miRNAs directly regulates the expression of p27 by targeting its 3′UTR sequence (Figure 4). *miR-221* and *miR-222*, both belonging to this miRNA family, were predicted and verified to downregulate p27 expression in cell culture models [159]. MYC plays a key role in the regulation of non-coding RNAs and thus, modulates the expression of their target genes, a mechanism that has become recently more evident. MYC-regulation of miRNA expression has been linked mainly with miRNAs targeting mRNAs involved in cell-cycle regulation. In fact, MYC directly regulates the *miR-221* family of miRNAs, which have been found to target p27 (and p57) [160]. Besides, *miR-221* and *miR-222* are consistently overexpressed in liver tumors, showing an opposite correlation with low levels of p27 due to its aberrant pros-transcriptional regulation. Furthermore, *miR-221* (but not *miR-222*) has been reported to enhance tumorigenesis not only in vitro, but also in vivo [161].

### 4.3. Sequestration of p27 by Cyclin D-CDK4/6 Complexes

Inhibition of the cell cycle by p27 is controlled, in a great extent, by its recruitment to cyclin D-CDK4/6 complexes. The shift of p27 from cyclin E/CDK2 to cyclin D-CDK4/6 complexes relieves cyclin E-CDK2 from p27-mediated inhibition allowing progression through the cell cycle [162]. Intriguingly, p27 binds constitutively to cyclin D-CDK4/6 complexes. Although considered a CDK inhibitor, p27 has been found associated with both, active and inactive cyclin D-CDK4/6 complexes, depending on the cell proliferation state and on the phosphorylated status of p27 [163,164]. In arrested cells, the unphosphorylated p27 impairs the activation of cyclin D-CDK4/6 complexes by blocking CDK-ATP binding pocket. However, upon mitogenic stimuli, p27 gets phosphorylated at Tyr74, Tyr88, and/or Tyr89, leading to a conformational change that releases the blockade of the ATP binding site and the CDK is further activated by the CAK [164]. Moreover, p27 (as well as p21) is known to stabilize these complexes, as p27 depletion leads to more unstable D-type cyclins and less cyclin D-CDK4/6 complexes. MYC induces the expression of D-type cyclins and CDK4 and CDK6 [13,103], thus leading to the formation of cyclin D-CDK4/6 complexes able to sequester p27 from cyclin E-CDK2. Activation of MYC in mouse cells containing the MYC-ER chimera, promoted the interaction of p27 with D-type cyclins in an extent that proportionally correlated with the levels of cyclin D induced by MYC and with the activation of cyclin E-CDK2 complexes [103]. Although it has been reported that MYC directly induces cyclins D1 and D2, there has been some controversy concerning cyclin D1 regulation. Different studies reported opposite effects in the regulation of cyclin D1 by MYC, depending on cell types and models used [103,165,166,167,168]. On the other hand, cyclin D2 is well known to be induced by MYC as recurrently reported [103,168,169,170]. Moreover, CDK4 is also a bona fide MYC target gene [171] which is activated by MYC, presumably through the E-boxes located along its promoter region. Indeed, MYC has been reported to induce CDK4 at the transcriptional level in human and rodent cells and it has been found to activate the CDK4 promoter in reporter assays [171]. Finally, CDK6 is induced by MYC at the mRNA level, although this induction does not correlate with CDK6 protein levels [172,173]. D-type cyclins and CDK4 and 6 are repressed by different miRNAs (as many other genes involved in cell-cycle progression) such as the let-7 family of miRNAs, *miR-34a*, *miR-15a/61*, and *miR-26a*. MYC has been reported to induce the expression of D-type cyclins and CDK4/6 by repressing these miRNAs. Altogether, MYC induces the formation of cyclin D-CDK4/6 complexes promoting the switch of p27 from cyclin E-CDK2 to cyclin D-CDK4/6 complexes, thus inducing the G1-S phase transition as reviewed in [13].

### 4.4. Induction of p27 Degradation Through the MYC/CDK2/SKP2 Axis

The most important regulation of p27 levels and thus, p27 activity, takes place in the nucleus and relies on p27 protein stability. Upon mitogenic stimuli, p27 levels within the cell need to be reduced to allow cyclin-CDK activation and cell-cycle progression. The most efficient way for the cell to overcome p27 inhibition is mediated by its degradation via proteasome. Proteasomal degradation of p27 is mainly mediated by the SCF^SKP2^ ubiquitin ligase complex [174,175,176] (Figure 5) which, as most of the SCF complexes, relies on a specific phosphorylation state of its target protein to be able to recognize and ubiquitylate it (reviewed in [177,178]). In the case of p27, phosphorylation at its Thr187 is essential for SCF^SKP2^ recognition [176]. p27 phosphorylation and subsequent degradation is induced by MYC, whereas mutation of the threonine of p27 at the position 187 impaired this effect [179,180]. Phosphorylation of the Thr187 of p27 is mainly mediated through cyclin E-CDK2 complexes, although it has been found to be phosphorylated as well by cyclin A-CDK2 and cyclin B-CDK1, although in a lesser extent and in vitro [181]. Moreover, cells lacking CDK2 have shown phosphorylation of p27 at the Thr187 residue, suggesting that, in the absence of CDK2, there are other/s kinases able to trigger this phosphorylation [182]. In the absence of CDK2, CDK4, and CDK6, the phosphorylation of p27 at Thr187 can be carried out by CDK1 [183]. In vivo, p27 phosphorylated at the Thr187 is found forming complexes with cyclin E/A-CDK2, but not with D-type cyclin complexes [180]. MYC activation of cyclin E-CDK2 complexes during G_1_ phase was first described in a Rat1-MYC-ER model [184] while the absence of MYC impaired cyclin E-CDK2 activation in exponentially growing conditions [185]. Moreover, *CCNE* was later reported to be a direct MYC target gene and that MYC could also induce its expression via E2F1, another MYC target gene needed for the G_1_ to S-phase transition [186,187]. In turn, some E2F factors (E2F1, 2, 3) can repress *MYC* whereas E2F7 transactivate *MYC* [188]. On the other hand, MYC directly represses certain miRNAs that target *CCNE*, such as *miR-34a* and *miR-26a* (reviewed in [13]. Thus, MYC activation of cyclin E-CDK2 complexes would rely mainly in MYC’s ability to induce cyclin E expression and form new and active cyclin E-CDK2 complexes. Nevertheless, activation of cyclin-CDK complexes is not only dependent on the regulatory subunit of the kinase (the cyclin), but also on the phosphorylation of certain residue within the CDK (Thr160 in CDK2 and their structurally equivalents in CDK1, CDK4, and CDK6). Phosphorylation of these residues is mediated by the CAK (CDK activating complex) [189,190,191], which consists of three subunits: cyclin H, CDK7 and MAT1. MYC increases CAK activity by augmenting the translation rates of the mRNA of its three components, leading to higher protein levels [192]. On the other hand, MYC activates CDK7 expression. MYC binds to its promoter sequence in mouse ES cells and CDK7 expression is reduced in MYC null rat cells [144,193], thus, MYC actively participates in the regulation of the CAK, promoting cyclin-CDK complexes activation and, in the case of cyclin E-CDK2 complexes, favoring phosphorylation and subsequent inactivation of p27 [194]. Phosphorylation of p27 by cyclin E-CDK2 led to p27 ubiquitination in vitro, suggesting that phosphorylated p27 was a target for the ubiquitin-proteasome degradation system while a p27 T187A mutant did not show this effect [181,195]. In fact, the F-box protein SKP2, which is part of an E3 ubiquitin ligase of the SCF complex, specifically recognizes p27 phosphorylated at the Thr187, promoting p27 degradation, and it is needed for the transition through quiescent state to S-phase. This process leads to the activation of cyclin A-CDK2 complexes inducing S-phase entry and DNA synthesis (Figure 5). Moreover, the T187A p27 mutant suppresses SKP2-induced cyclin A activation and S-phase entry [176,196,197].

The SCF^SKP2^ complex is composed by RBX1, CUL1, SKP1, and the F-box protein SKP2 [177,178]. Cell-free extract assays revealed that SKP2 binds to phosphorylated p27 at the C-terminal domain, while the lack of that phosphorylation totally abolished the interaction. Immunodepletion of CUL1, SKP1, or SKP2 abolished p27 degradation [176]. Unlike any other SCF substrate, p27 ubiquitination requires the accessory protein CKS1, which appears to be necessary to bridge between p27 and SCF^SKP2^. The N-terminal portion of p27 packs with SKP2, a central Glu^185^ side chain inserts between SKP2 and CKS1 and the C-terminal portion containing the phosphorylated Thr187 binds to CKS1 [198]. Further, cyclin A-CDK2 complexes facilitate the recruitment of p27 to the SCF^SKP2^-CKS1 stimulating p27 ubiquitination [181,199,200]. Cyclin A interacts with SKP2 while CKS1 does with CDK2 and both interactions are essential, as disruption of any of them abolished p27 ubiquitination [200,201,202].

MYC induces p27 proteasomal degradation through the upregulation of the SCF^SKP2^ complex. This is achieved as several components of this complex have been reported to be MYC-target genes: CUL1, CKS1, and SKP2 [179,203,204]. While CKS1 is indirectly induced by MYC, most likely through other transcription factors regulated by MYC and involved in CKS1 transcriptional regulation [203], CUL1 and SKP2 have been described to be direct-MYC target genes. Activation of MYC-ER by 4-hydroxy-tamoxifen resulted in increased mRNA levels of CUL1 and SKP2, even in the absence of de novo protein synthesis. Both contain canonical E-boxes within their core promote and have been reported to be essential for MYC-transcriptional regulation of these genes [179,204]. Depletion of any of the three (CUL1, CKS1, and SKP2) lead to increased p27 protein levels and arrested proliferation, and MYC was unable to counteract it. Besides, overexpression of CUL1 or CKS1 in null *MYC* cells, which reduced p27 levels within the cells, and siRNA mediated depletion of p27, restored MYC’s wild type phenotype leading to normal proliferation rates. Thus, the SCF^SKP2^ complex is essential for MYC’s activity as a pro-proliferative transcription factor, by means of reducing p27 levels to allow cell-cycle progression. Altogether, it brings to light that MYC plays a critical role in the regulation of p27 degradation via proteasome through the combination of different regulatory mechanisms.

## 5. MYC-Mediated Synthetic Lethality and the Cell Cycle

MYC would be a good target for therapy. First, MYC deregulation occurs frequently in human cancer. Second, MYC addiction has been shown in several models so that inactivation or depletion of MYC leads to tumor regression [205,206,207]. The oncogene addiction is defined as the phenomenon by which some tumors exhibit a dependence on a single oncogenic protein or pathway for sustaining growth and proliferation [208].

Third, whole-body inactivation of MYC in mouse models by expression of a dominant negative MYC form (Omomyc, a peptide that interferes with MYC-MAX interaction [209]) only provokes mild side effects. This suggest that pharmacological inhibition of MYC could likely be implemented without major side effects [210,211]. However, to date no anti-MYC drugs have reached the clinical use. Like other transcription factors, MYC has the reputation of a non-druggable target. Despite that, several approaches have targeted MYC. Inhibitors of bromodomain BRD4 protein (JQ1, OTX15 and derivatives) that repress MYC expression [212,213] (Figure 6a) have been tested in clinical assays in lymphoma, but the drug is not specific for MYC but also represses other genes which transcription is dependent on BRD4 [214].

Several molecules have been described to bind MYC and impair its function. Most of these molecules interrupt the MYC-MAX interaction (Figure 6b), as the peptide Omomyc does [215]. Most of these compounds, i.e., 10058-F4 and 10074-G5, were discovered using a two-hybrid system [216]. These inhibitors are specific for MYC and have been broadly used in preclinical studies, but they have not reached the clinical use due to its low potency and to its rapid degradation [217].

A more promising approach is to target MYC as an indirect target via synthetic lethal approaches (Figure 6c). Several putative synthetic lethal genes have been identified [218,219], including CDKs. Indeed, the first synthetic lethal MYC interactor described was CDK2, and most of the synthetic lethal combinations of MYC so far reported involve enzymes that functions in cell cycle. They will be briefly discussed below.

### 5.1. MYC and CDK1 Inhibitors

CDK1 is essential for mammalian cell division [220] and is the only CDK required for completion of cell cycle in animal cells [221]. A number of small molecule inhibitors of CDK1 have been developed. Most of them induce an arrest in G_2_ phase, and some are being used in clinical trials [222,223]. Accordingly, a CDK1 inhibitor induces cell death in Burkitt lymphoma and multiple myeloma cell lines depending on MYC levels, and CDK1 inhibition in Eμ-Myc mice results in extended survival [224]. Similar observations were made in breast cancer cells [225]. These results suggest that CDK1 inhibition is synthetic lethal on MYC expressing cells. However, purvalanol A is selective but not specific for CDK1, and shows some activity against other CDKs [222]. Therefore, the possibility existed that other kinases could contribute to the synthetic lethal effect. However, we have recently shown, using genetic approaches, that CDK1 inhibition is enough for the synthetic lethality with MYC in mouse embryo fibroblasts as it occurs in cells deficient in CDK2, CDK4, and CDK6 [183]. It is worthy to note that CDK1 not only arrests the cell cycle but also plays a role in DNA replication and DNA repair [226]. On the other hand, MYC-induced carcinogenesis is associated to genomic instability, as demonstrated in cell culture and in mice models (reviewed in [227,228]). MYC impairs DNA repair [229] and induces unscheduled DNA replication [230,231,232,233,234,235]. Therefore, it is conceivable that not only cell-cycle arrest but also the impairment of DNA repair is part of the molecular mechanism of the synthetic lethality between MYC overexpression and CDK1 inhibition.

### 5.2. MYC and Aurora Kinase Inhibitors

Aurora kinases A and B (AURKA and AURKB) are serine/threonine kinases required for mitosis [236]. MYC regulates Aurora Kinase A [237]. Expression of MYC but not that of other oncogenes, made the cells much more sensitive to Aurora kinase inhibitors (e.g., AS703569), being AURKB the central target in this model. Another Aurora kinase inhibitor, VX-680, was demonstrated to selectively kill the cells that overexpress MYC [238]. Indeed, MYC expression levels may provide a biomarker to identify tumors that may respond to aurora B kinase inhibitors. Moreover, the drug inhibited AURKB in vivo using mouse models that develop either B-cell or T-cell lymphomas in response to MYC overexpression, and the lethal response is independent of p53-p21 pathway [239]. This fact is relevant since *TP53* is frequently mutated in cancer and usually confers an adverse prognosis.

### 5.3. MYC and CHK1 Inhibitors

One of the effects of MYC overexpression is to induce DNA replicative stress [13], which in turn activates CHK1 (Checkpoint Kinase 1). CHK1 is a serine/threonine kinase that functions as a major component of the DNA damage response. CHK1 regulates cell-cycle checkpoints following genotoxic stress to prevent the entry of cells with damaged DNA into mitosis and coordinates various aspects of DNA repair, and a number of molecules have been described as CHK1 inhibitors [240,241]. In cells from human and murine B-cell lymphomas there is a correlation between MYC and CHK1 levels, although CHK1 seems to be an indirect target of MYC [242]. Silencing of CHK1 with siRNA technology or inactivation with a small molecule results in selective death of MYC-overexpressing cells. These evidences turned CHK1 into an attractive therapeutic target. A CHK1 inhibitor (Chekin), was tested in the λ-Myc mouse model, where MYC induces lymphomas. In this model CHK1 inhibition was able to induce a significantly slower disease progression [242].

### 5.4. MYC and CDK9 Inhibition

CDK9 is not a kinase involved in cell-cycle progression but in transcription initiation. However, it is worth noting this interaction, given its similarity with cell cycle CDKs. Inhibition or depletion of CDK9 (with shRNAs) in cells and mouse models of hepatocellular carcinoma, results in delay of growth and the extent of its effect correlates with MYC levels, suggesting a synthetic lethal inhibition [243].

## 6. Concluding Remarks

The importance in cancer research of the set of proteins acting as physiological brakes of the cell cycle has been well established. On the other hand, the impairment of CKI activities is a major mechanism for the tumorigenic effects of MYC. Therefore, the deciphering of the molecular clues of the mechanisms leading to MYC-mediated inhibition of p21, p27, and p15 functions or expression is critical for the design of therapeutic approaches of cancers with MYC deregulation.

## Figures and Tables

**Figure 1 genes-10-00244-f001:**
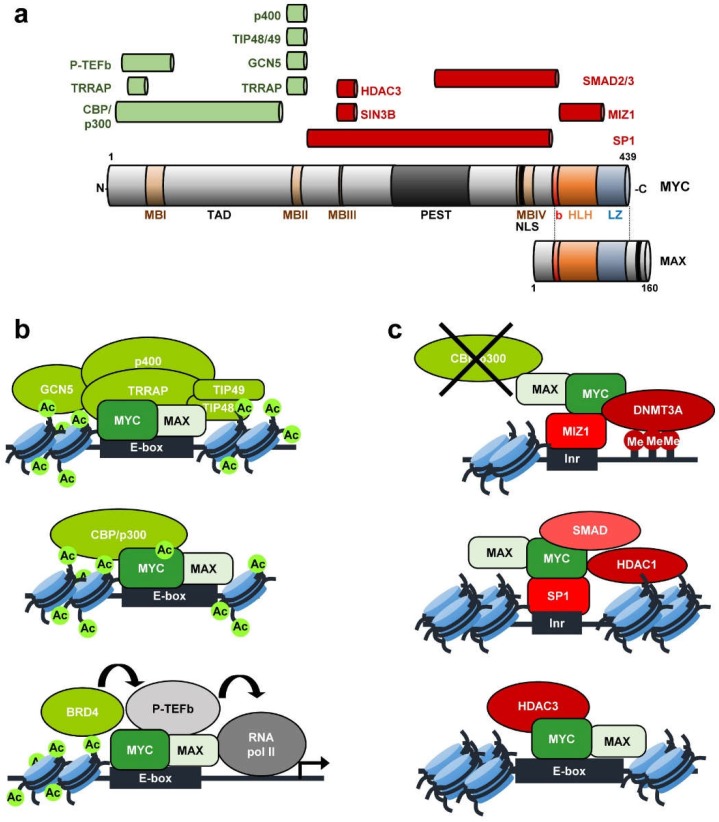
Oncogene c-MYC (MYC) structure and interaction complexes. (**a**) MYC structural domains are represented. MB, MYC boxes I-IV; TAD, transactivation domain; PEST, PEST sequence; NLS, Nuclear Location Signal. b, basic; HLH, Helix-Loop-Helix; LZ, Leucine Zipper. Through these domains, MYC interacts with different cofactors involved in transcriptional activation (in green) or repression (in red). MYC-MAX interaction is also indicated. (**b**) Transcriptional activation through MYC-associated complexes. Upper: MYC-MAX heterodimers bind E-box sequences and interact with co-activators such as TRRAP, GCN5 and others. These complexes mediate histone acetylation to transactivate MYC target genes. Middle: CBP/p300 also mediates MYC acetylation and increased stability. Bottom: BRD4 is a reader of acetylated histones and promotes the activity of P-TEFb complex, composed of CyclinT1 and CDK9. MYC interacts with P-TEFb, which phosphorylates the C-terminal domain of RNA polymerase II to trigger elongation. (**c**) Transcriptional repression through MYC-associated complexes. Upper: MYC interacts with MIZ-1, displacing coactivators with HAT activity such as CBP/p300. The MYC/MIZ-1 complex binds to Initiator element (Inr) sequences and recruits the DNA methyltransferase DNMT3A to repress transcription. Middle: SP1-SMAD complex is repressed by MYC. Recruitment of HDAC1 contribute to histone deacetylation nearby Inr sequences. Bottom: MYC also recruits HDAC3 to E-box sequences, reducing histone acetylation.

**Figure 2 genes-10-00244-f002:**
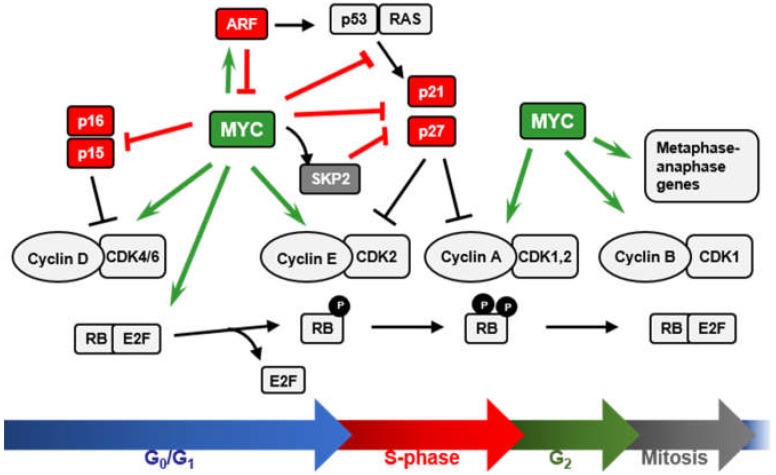
Impact of MYC on cell-cycle regulation. MYC stimulates cell-cycle progression and the cellular proliferation through the regulation of genes related to cell-cycle control. MYC induces positive cell-cycle regulators such as several cyclins, CDKs and E2F transcription factors (green arrows). Cyclin-CDK complexes phosphorylate RB, releasing E2Fs from the inhibitory interaction with RB, and allowing the expression of E2F target genes and the progression through the cell-cycle phases. MYC also represses genes encoding cell-cycle inhibitors such as p15, p21, or p27 (red bars), by different mechanisms. The regulatory mechanisms by which MYC antagonizes the activity of cell-cycle inhibitors are detailed in the text.

**Figure 3 genes-10-00244-f003:**
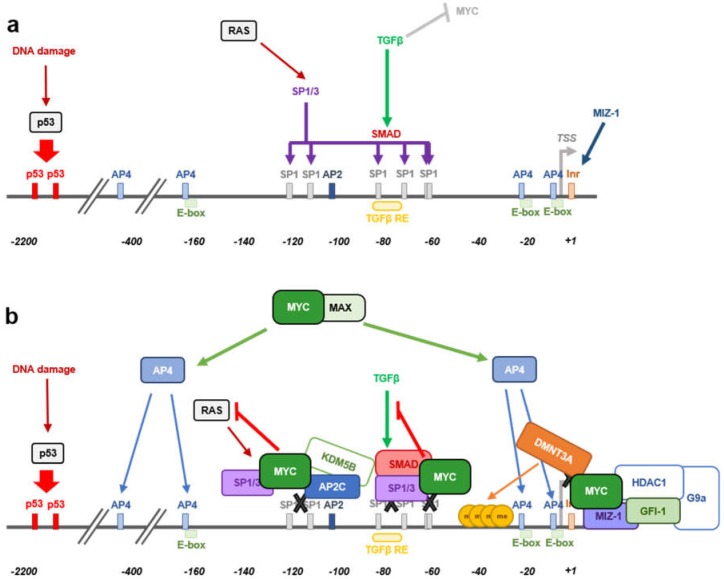
Transcriptional repression of *CDKN1A* (p21) by MYC. (**a**) Scheme of *CDKN1A* promoter region showing the localization of responsive elements for different transcription factors. In response to DNA damage, p53 induces *CDKN1A* transcriptional activation upon binding to the indicated sites. RAS mediates p21 induction through the SP1 sites. TGFβ also induces p21 and cell-cycle arrest upon binding to TGFβ-responsive element in the promoter. MIZ-1 transcription factor binds to the initiator sequence (Inr) to trigger p21 expression. TSS, transcription start site. (**b**) MYC represses *CDKN1A* at different levels. MYC is recruited to the promoter by interacting with SP1 and MIZ-1. At the Inr, MYC interacts with MIZ-1 and DNMT3A or forms a ternary complex with MIZ-1 and GFI-1, recruiting HDAC1 and G9a to repress p21. MYC interaction with SP1 counteracts SP1 induction of p21 by RAS or TGFβ signal. AP4 is a direct MYC target gene which mediates *CDKN1A* repression upon binding to the indicated sites in the promoter. MYC interacts with AP2C and recruits KDM5B through the AP-2 binding site promoting histone demethylation and *CDKN1A* repression.

**Figure 4 genes-10-00244-f004:**
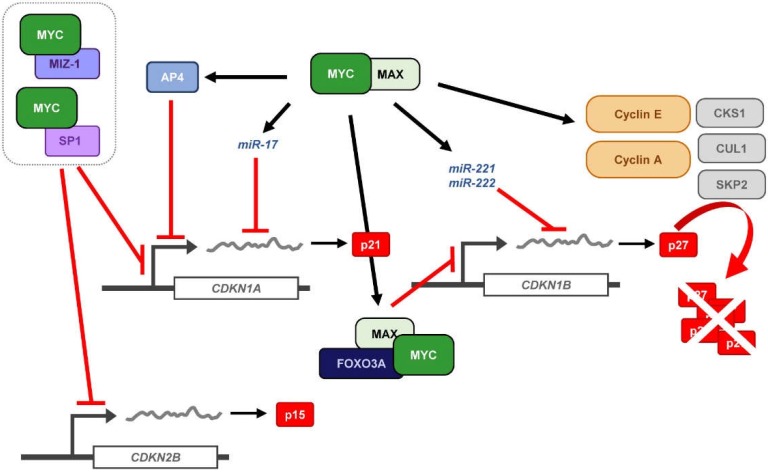
Mechanisms for MYC-mediated antagonism of CKIs p15, p21, and p27. MYC directly represses *CDKN2B* and *CDKN1A* promoters by interacting with MIZ-1 or SP1. MYC also induces the transcriptional repressor AP4 or the microRNA mir-17 to repress p21 expression. Although MYC-mediated repression of *CDKN1B* has been reported, the major mechanisms for the abrogation of p27 are related to the sequestering of p27 by cyclin-CDK complexes and to p27 degradation. See text for details.

**Figure 5 genes-10-00244-f005:**
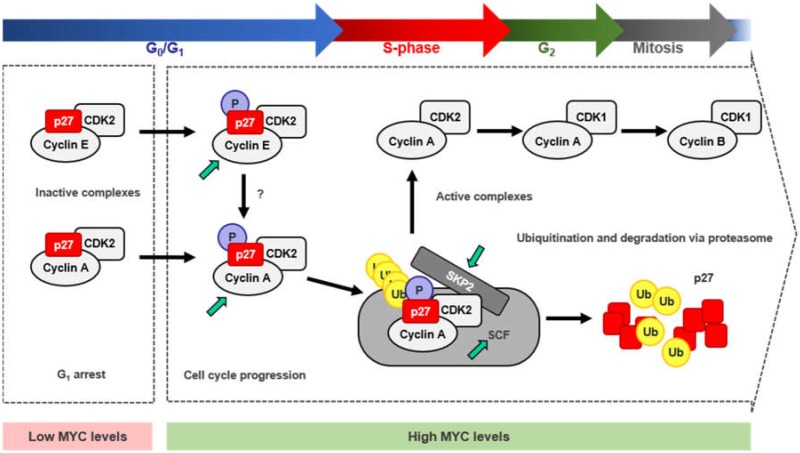
Mechanism for p27 degradation through MYC/CDK2/SKP2 axis. Scheme summarizing the proteasomal degradation of p27 by the SCF^SKP2^ complex induced by MYC. During arrested cell proliferation and low MYC levels, unphosphorylated p27 binds and inhibits cyclin E/A-CDK2 complexes. MYC-mediated transition from G1 to S-phase, involves the activation of Cyclin E/A-CDK2 complexes and the phosphorylation of p27 at the Thr187. Phosphorylated p27 is recognized by the SCF^SKP2^ ubiquitin-ligase complex when bound to cyclin A-CDK2 complexes and is ubiquitinated and targeted for degradation via proteasome. This process releases the inhibition of Cyclin-CDK complexes by p27 allowing the transition along the different phases of the cell cycle promoting proliferation. Green arrows show MYC target genes.

**Figure 6 genes-10-00244-f006:**
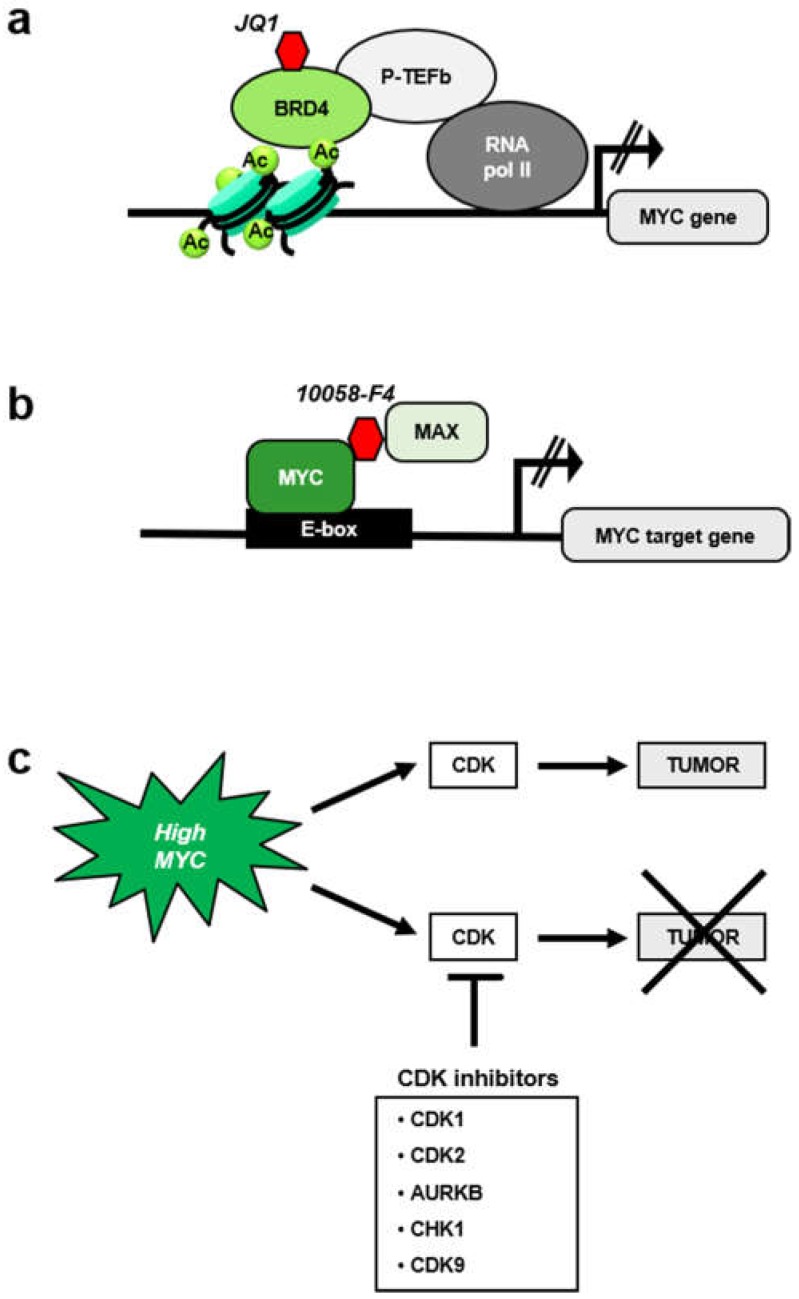
MYC as a therapeutic target in cancer. (**a**) Mechanism of action of BRD4 inhibitors, such as JQ1, as anti-MYC drugs. (**b**) MYC inhibition by blocking MYC–MAX interaction with small molecules, such as 10058-F4. (**c**) Scheme for MYC-mediated synthetic lethality with inhibitors of kinases (CDKs and others) involved in cell-cycle regulation.

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
