# Peer review of "MYC Oncogene Contributions to Release of Cell Cycle Brakes"

_genes, 2019, doi:10.3390/genes10030244_

Round 1

Reviewer 1 Report

The review article “MYC oncogene as the cell-cycle brakes releaser” by Lucía García-Gutiérrez and colleagues is a very extensive and comprehensive analysis of the literature describing the role of the oncogene MYC in cell cycle. Furthermore, by focusing mainly on cell cycle inhibitors the authors add a different point of view to this extensively reviewed topic.

Few points need to be addressed before publication:

Page 2, line 45: genomic instability is listed as a biological function deregulated by MYC: I would not say that genomic instability is a biological function. I would rather write that MYC deregulate DNA repair or genome maintenance. The same is for differentiation impairment: I would rather omit impairment.

Page 5, line 208: the sentence: “....resulted in the inhibition of p15 mediated by TGFβ” sounds as the inhibition of p15 is mediated by TGFβ. I would rather write: “... resulted in the inhibition of TGFb-mediated p15 induction”.

Page 5, line 214: I would change the sentence “To do so, different MYC partners have been described: YY1, TFII-I and MIZ-1” with: “Different proteins have been described to cooperate with MYC in the binding to the Inr element.”

Page 5, line 225: remove “among others”

Page 6, line 249: I would remove “..like those of p53null lymphomas”. It sounds like p53 null lymphomas have wild-type p53...

Page 6, line 260: I would change the sentence “... although this has not been found that evident in a reported mouse model” with “... although this pathway does not seems to be conserved in mouse”.

Page 7, line 281: the authors mention MYCER for the first time without specifying what it is: is this MYC-ER? if yes the sentence written later, in line 309, should be moved here.

Page 8, line 341: I would write that a sequence is short, rather than small.

Page 13, line 528: “CDKN1B mRNA expression levels opposite correlate with MYC expression” should be “CDKN1B mRNA expression levels inversely correlate with MYC expression”

Line 536: remove the second “which”.

Line 539: the MYC P115L mutant has not been described before in the text.

Line 564: Is not clear what the Authors mean with displacement.

Line 587: The concept that Myc regulates the expression of several miRNAs has been already stressed before. I would therefore remove here the sentence: “MYC is also known to play a key role in the regulation of miRNAs”

Author Response

Response to Reviewer 1 Comments

Point 1: Page 2, line 45: genomic instability is listed as a biological function deregulated by MYC: I would not say that genomic instability is a biological function. I would rather write that MYC deregulate DNA repair or genome maintenance. The same is for differentiation impairment: I would rather omit impairment.

Response 1: We agree with the reviewer regarding to the fact that “genome instability” is not a biological function and so the statement was not properly written. We have replaced “genome instability” by “genome maintenance”. In the same way we have omitted “impairment”.

Point 2: Page 5, line 208: the sentence: “....resulted in the inhibition of p15 mediated by TGFβ” sounds as the inhibition of p15 is mediated by TGFβ. I would rather write: “... resulted in the inhibition of TGFb-mediated p15 induction”.

Response 2: We agree with the reviewer and we have changed the text so that it states as follows: “However, exogenous MYC expression resulted in the inhibition of TGFβ-mediated p15 induction”.

Point 3: Page 5, line 214: I would change the sentence “To do so, different MYC partners have been described: YY1, TFII-I and MIZ-1” with: “Different proteins have been described to cooperate with MYC in the binding to the Inr element.”

Response 3: In agreement with the reviewer suggestion, we have corrected the text and it states as follows: “Different proteins have been described to cooperate with MYC in the binding to the Inr element, such as YY1, TFII-I and MIZ-1”

Point 4: Page 5, line 225: remove “among others”

Response 4: “among others” has been removed.

Point 5: Page 6, line 249: I would remove “..like those of p53null lymphomas”. It sounds like p53 null lymphomas have wild-type p53...

Response 5: We agree with the reviewer that the way this sentence is written can lead to misunderstanding. In this case, we wanted to emphasise that Eµ-myc mice lacking ARF showed a phenotype similar to p53-null animals, despite harbouring wild type p53. We have re-written the text to make it clearer: “In agreement, INK4A/ARF-/--Eµ-MYC mice were more prone to develop lymphomas and displayed apoptotic defects despite the presence of wild-type p53, a phenotype similar to the one observed in p53-null lymphomas.”

Point 6: Page 6, line 260: I would change the sentence “... although this has not been found that evident in a reported mouse model” with “... although this pathway does not seems to be conserved in mouse”.

Response 6: In agreement with the reviewer, we have changed the sentence and it states as follows: “On the other hand, the transcription factor E2F1 directly induces ARF [84], although this pathway does not seem to be conserved in mouse.”

Point 7: Page 7, line 281: the authors mention MYCER for the first time without specifying what it is: is this MYC-ER? if yes the sentence written later, in line 309, should be moved here.

Response 7: Yes, MYCER is MYC-ER (this has been changed in the text). The reviewer comment is right. The explanation of what this fusion protein is has been moved here and removed from line 309: “In contrast, other studies have shown that upon ectopic MYC expression, ARF is relocalized from the nucleolus to the nucleoplasm and colocalized with it. The same result was obtained upon MYC-ER activation, a chimeric protein consisting of MYC fused to the estrogen receptor and activatable by 4-hydroxy-tamoxifen.”

Point 8: Page 8, line 341: I would write that a sequence is short, rather than small.

Response 8: We have replaced “small” for “short” in the text as suggested.

Point 9: Page 13, line 528: “CDKN1B mRNA expression levels opposite correlate with MYC expression” should be “CDKN1B mRNA expression levels inversely correlate with MYC expression”

Response 9: We have changed “opposite” for “inversely” in the text as suggested.

Point 10: Line 536: remove the second “which”.

Response 10: The second “which” has been removed from the text.

Point 11: Line 539: the MYC P115L mutant has not been described before in the text.

Response 11: The description of the MYC P115L mutant has now been included in the text as indicated by the reviewer: “MYC interaction with the Inr element relies on its MBII and accordingly, a MYCP115L mutant, in which the Phe at the position 155 is replaced by a Leu within the region of MYC needed for its transcriptional suppression function, enhances its repressor activity [152], consistently with other known MYC-repression mechanisms”

Point 12: Line 564: Is not clear what the Authors mean with displacement.

Response 12: With “displacement” we meant the shift of p27 from cyclin E/CDK2 to cyclin D-CDK4/6 complexes. The sentence has been re-written to better clarify this point and now it states as follows: “The shift of p27 from cyclin E/CDK2 to cyclin D-CDK4/6 complexes relieves cyclin E-CDK2 from p27-mediated inhibition allowing progression through the cell cycle.”

Point 13: Line 587: The concept that Myc regulates the expression of several miRNAs has been already stressed before. I would therefore remove here the sentence: “MYC is also known to play a key role in the regulation of miRNAs”

Response 13: We agree with the reviewer and the sentence “MYC is also known to play a key role in the regulation of miRNAS” has been removed.

Reviewer 2 Report

This review, on the relationship of MYC function to cell cycle stimulation through abrogation of cell cycle inhibitors, is carefully researched, well organized and comprehensive (perhaps overly so). It will likely serve as a valuable resource for those interested in the regulation of cell proliferation as well as for researchers in the MYC/oncogenesis fields. The many figures are especially useful in laying out the molecular interactions and the multiplicity of protein complexes involved.

While the review is well written overall there are several places where the English could be improved. Perhaps most noticeable is the title itself: “MYC oncogene as the cell-cycle brakes releaser” is awkward and implies that MYC is the only way to release a cell cycle brake. A suggestion: The MYC oncogene mediates cell cycle brake release. 

Also on line 25: “MYC here in after” should be: referred to herein as MYC.

Author Response

Response to Reviewer 2 Comments

Point 1: While the review is well written overall there are several places where the English could be improved. Perhaps most noticeable is the title itself: “MYC oncogene as the cell-cycle brakes releaser” is awkward and implies that MYC is the only way to release a cell cycle brake. A suggestion: The MYC oncogene mediates cell cycle brake release.

Response 1: We agree with the reviewer regarding to the fact that the tittle can be confusing. Following his/her advice, we have changed the tittle for the following one:

“MYC oncogene contributions to release of cell cycle brakes”

Point 2: Also on line 25: “MYC here in after” should be referred to herein as MYC.

Response 2: As indicated by the reviewer, the statement “MYC here in after” has been replaced by “referred to herein as MYC”.
